# Non-Contact Fatigue Estimation in Healthy Individuals Using Azure Kinect: Contribution of Multiple Kinematic Features

**DOI:** 10.3390/s25216633

**Published:** 2025-10-29

**Authors:** Takafumi Yamada, Kai Kondo

**Affiliations:** 1National Institute of Technology, Tsuyama College, 624-1 Numa, Tsuyama 708-8509, Japan; 2Kyoto Seisakusho Co., Ltd., 377-1 Yodomizu-cho, Fushimi-ku, Kyoto 613-0916, Japan

**Keywords:** fatigue estimation, Azure Kinect, non-contact sensing, exercise monitoring, sports engineering, healthcare applications

## Abstract

Monitoring exercise-induced fatigue is important for maintaining the effectiveness of training and preventing injury. We evaluated a non-contact approach that estimates perceived fatigue from full-body kinematics captured by an Azure Kinect depth camera. Ten healthy young adults repeatedly performed simple, reproducible whole-body movements, and 3D skeletal coordinates from 32 joints were recorded. After smoothing, 24 kinematic features (joint angles, angular velocities, and cycle timing) were extracted. Fatigue labels (Low, Medium, and High) were obtained using the Borg CR10 scale at 30-s intervals. A random forest classifier was trained and evaluated with leave-one-subject-out cross-validation, and class imbalance was addressed by comparing no correction, class weighting, and random oversampling within the training folds. The model discriminated fatigue levels with high performance (overall accuracy 86%; macro ROC AUC 0.98 (LOSO point estimate) under oversampling), and feature importance analysis indicated distributed contributions across feature categories. These results suggest that simple camera-based kinematic analysis can feasibly estimate perceived fatigue during basic movements. Future work will expand the cohort, diversify tasks, and integrate physiological signals to improve generalization and provide segment-level interpretability.

## 1. Introduction

Regular physical activity is widely recommended to extend healthy life expectancy and prevent lifestyle-related diseases [1]. However, training that is too light has limited benefits, whereas excessive intensity can lead to accumulated fatigue and a higher risk of injury [2]. Consequently, there is a growing need for objective, timely monitoring of exercise-induced fatigue to provide individualized feedback on appropriate training intensity.

Existing approaches often rely on wearable or physiological sensors such as heart-rate/heart-rate variability (HR/HRV) monitors, inertial measurement units (IMUs), and electromyography (EMG) [3,4,5]. Although conventional wearable approaches are effective, body attachment can interfere with natural movement. By contrast, recent microwearables (e.g., skin-interfaced microfluidics and ultrathin bioelectronics) are designed to be skin-conformal and low burden, with wireless communication, thereby substantially reducing motion interference [6,7]. Nevertheless, many platforms remain at the prototype stage, and their cost and availability are not yet competitive with commodity, image-based, non-contact technologies. In this study, we position a single-camera, skeleton-based pipeline as a camera-only reference baseline that complements microwearables and multimodal systems (see Section 5).

In parallel, advances in depth cameras and image-based body tracking (e.g., Microsoft Azure Kinect) have enabled non-contact acquisition of full-body kinematics for motion analysis and fatigue assessment [8,9,10]. Prior studies suggest that vision-based sensing can provide unobtrusive measures of fatigue; however, feasibility and reliability remain underexplored for simple, reproducible exercises performed by healthy individuals, especially in small, class-imbalanced cohorts.

Against this background, we investigated whether perceived fatigue can be estimated from non-contact full-body kinematics captured by an Azure Kinect depth camera and analyzed using a standard machine learning pipeline. To improve reproducibility and generalizability, we used a simple, familiar whole-body movement that induces fatigue quickly and is reliably tracked by Kinect. Perceived fatigue was labeled with the Borg CR10 and categorized into three levels (Low, Medium, High; abbreviated L/M/H in figures and tables). We chose CR10 for its practicality in short trials. We also acknowledge alternative criteria (e.g., physiological measures, talk test) and discuss their implications in Section 5.

Our contributions are threefold:A feasibility evaluation of non-contact fatigue estimation in healthy young adults performing a reproducible movement;The definition of 24 kinematic features from 32 joints;An assessment of class imbalance handling strategies and their impact on performance.

## 2. Related Work

Sensor-based monitoring of exercise-induced fatigue has been extensively studied across multiple modalities. Conventional approaches rely on contact-based, physiological, and biomechanical sensors. For example, Shaffer and Ginsberg [2] summarized heart rate variability metrics as widely used indicators of physical strain. Gualtieri et al. [3] and Patel et al. [5] reviewed wearable sensing for biomechanics and rehabilitation, emphasizing its applicability to daily activity monitoring and practical deployment. Chowdhury et al. [4] discussed surface EMG, which provides detailed muscular fatigue information, but requires careful electrode placement and signal processing. While effective, these wearable and physiological approaches demand continuous skin contact or specialized preparation, limiting their practicality in everyday exercise scenarios.

In contrast, vision-based non-contact sensing has emerged as a promising alternative. Shotton et al. [8] introduced a real-time method for human pose recognition from depth images, establishing a foundation for skeleton tracking using Kinect. Clark et al. [9] validated Kinect for postural control assessment, whereas Pfister et al. [10] compared Kinect with marker-based motion capture for gait analysis, confirming reasonable accuracy for many biomechanical applications. These studies indicate that Kinect-based kinematic measurements can be acquired unobtrusively and at low cost.

Several studies have extended Kinect to fatigue-monitoring applications. Albert and Arnrich [11] proposed a computer vision approach to continuously monitor fatigue during resistance training. Zhao et al. [12] developed a Kinect-based system to measure occupational fatigue in workers, demonstrating its feasibility outside of laboratory settings. Wakimoto et al. [13] applied Laban movement analysis with Kinect to estimate fatigue based on expressive body motion descriptors. While these studies demonstrate the versatility of Kinect-based sensing, they primarily target resistance exercises, workplace ergonomics, or movement analysis frameworks. Moreover, many existing studies have focused on patients or rehabilitation populations, leaving the potential for simple, reproducible exercises in healthy individuals.

In summary, prior research has shown that Kinect provides reliable kinematic data and has been applied to fatigue estimation. However, the feasibility of classifying fatigue in healthy young adults performing simple, familiar whole-body movements, and the role of feature patterns across multiple categories in such classifications, remain open questions. To address this gap, we investigated whether perceived fatigue can be estimated from non-contact full-body kinematics captured by an Azure Kinect depth camera and analyzed using a standard machine learning pipeline.

## 3. Materials and Methods

### 3.1. Participants

Ten healthy young adults (eight males, two females) participated in this study. None reported any musculoskeletal or neurological disorders. The study was approved by the Ethics Committee of the National Institute of Technology, Tsuyama College (approval No. 2023-03). All participants were briefed on the study purpose, procedures, potential risks (transient fatigue), confidentiality, and data handling; participation was voluntary, and they could discontinue at any time without penalty. Written informed consent was obtained prior to data collection. Participant characteristics are summarized in Table 1 (mean ± SD; ranges); anthropometric measurements (age, height, mass, BMI) were recorded for all ten participants, and no imputation was required.

Accordingly, we adopted a small pilot cohort (*n* = 10), consistent with guidance for feasibility studies in which ≈10–12 participants are commonly used to assess procedures and obtain preliminary variance/effect-size estimates for subsequent sample-size calculations, rather than to test hypotheses [14,15,16]. Accordingly, we adopted a small pilot cohort (*n* = 10), consistent with common practice in feasibility studies; we acknowledge the limited generalizability and will use these estimates to plan a larger, sex-balanced follow-up with an a priori sample-size calculation.

### 3.2. Experimental Protocol and Labeling

A standardized instruction was given: participants alternated between a neutral stance and an arms-abducted posture at a comfortable, self-selected pace for up to 3 min, without external pacing or feedback. Borg CR10 ratings were verbally reported every 30 s. An investigator was present throughout to ensure safety and to terminate the task on request or if any discomfort occurred. Figure 1 illustrates the repeated movement task (start and spread postures) and stance width (≈0.80 m). We adopted the Borg CR10 to reflect perceived fatigue because it is widely used in exercise settings, requires no additional hardware, and can be administered repeatedly during short trials without interrupting the task. We prioritized a device-free protocol that would not interrupt short in-task windows; therefore, a talk test was not included at this feasibility stage. The participants could discontinue earlier if desired. A standardized instruction was given: participants alternated between a neutral stance and an arms-abducted posture at a comfortable, self-selected pace for up to 3 min, without external pacing or feedback. Borg CR10 ratings were verbally reported every 30 s. An investigator was present throughout to ensure safety and to terminate the task on request or if any discomfort occurred.

For the analysis, each CR10 report was used to label the immediately preceding 30-s movement segment with three levels: Low (CR10 0–3), Medium (CR10 4–6), and High (CR10 ≥ 7). Labels were derived exclusively from CR10; no talk test or speech-based criteria were administered in this feasibility study, and no additional physiological measures were used for labeling. Alternative criteria include heart rate (HR) and heart rate variability (HRV), surface electromyography (EMG; e.g., median-frequency shifts), blood lactate, and oxygen-consumption-based measures, as well as the talk test. These were intentionally not used here to minimize burden in a camera-only feasibility study and to align labels with subjective experience. The advantages and limitations of CR10-only labeling and our plan to integrate objective checks (e.g., HR/HRV, surface EMG, talk test) are discussed in Section 5.

### 3.3. Data Acquisition (RGB-D Capture and Body Tracking)

Full-body kinematic data were captured as RGB-D (color + depth) frames using the Azure Kinect DK (Microsoft, Redmond, WA, USA). RGB-D frames were recorded with the Azure Kinect Sensor SDK and processed with the Azure Kinect Body Tracking SDK (v1.1.x) to estimate 3D positions of 32 anatomical joints using a deep learning model [17,18,19,20]. The camera was mounted 1.15 m above the floor and positioned 1.47 m in front of the participant (frontal view) to minimize occlusion (Figure 2a). Joint naming and hierarchy followed the Azure Kinect Body Tracking SDK (Figure 2b). To illustrate the tracking output, Figure 2c shows an example frame with the 32-joint skeleton overlaid on the depth image, and Appendix A provides a short clip presented sequentially—RGB, then depth, then depth with skeleton overlay (≈5 s each).

Joint coordinates returned by Azure Kinect Body Tracking SDK are expressed in millimeters in the depth-camera coordinate system; coordinates were converted to meters (divided by 1000) prior to computing angles and velocities. The time series were smoothed before differentiation to reduce high-frequency noise (Section 3.4). The timing of the Borg CR10 reports (every 30 s) and label assignments are described in Section 3.2.

### 3.4. Feature Extraction

We adopted a feature-level representation because fatigue manifests as subtle coordination and timing shifts across joints rather than the on/off presence of a coarse action label. In this feasibility stage, we deliberately chose a single, simple cyclic movement instead of multiple “normal actions” (e.g., jumping, running, squatting) to control task-related variability and safety burden and to ensure reproducibility under LOSO in a small cohort. Fast, high-impact actions introduce larger inter-subject variability and occasional tracking dropouts/occlusion in depth-based skeleton tracking; a controlled task allowed us to isolate fatigue-related changes.

Accordingly, we defined 24 kinematic features spanning three categories—joint angles, angular velocities, and cycle timing—as a sensitive, task-agnostic representation for fatigue estimation. Joint coordinates returned by the Azure Kinect Body Tracking SDK are expressed in millimeters in the depth-camera coordinate system; we converted them to meters (divided by 1000) before computing angles and velocities. The time series were smoothed before differentiation to reduce high-frequency noise. Table 2 lists all feature definitions.

Features were computed on a per-cycle basis; unless stated otherwise, results aggregate three consecutive cycles by averaging to reduce variance. Windows were non-overlapping, and Borg CR10 labels recorded every 30 s were aligned to the nearest timestamp. Figure 3 summarizes the end-to-end processing pipeline—from RGB-D capture and body tracking to preprocessing, feature extraction, and classification/evaluation. Note that the 24 kinematic features constitute the model input representation; evaluation is based on classification metrics—accuracy and macro ROC AUC (one-vs-rest)—reported under LOSO.

### 3.5. Classification and Evaluation

We trained a random forest (RF) and evaluated it with leave-one-subject-out cross-validation (LOSO). We preferred RF over a single decision tree because, in this small-cohort LOSO setting, single-tree models exhibited fold-to-fold instability (high variance). RF reduces variance through bootstrap aggregation and random feature subsampling (here, max_features = “sqrt”), providing more stable generalization while retaining straightforward interpretability via feature importances. In each fold, one subject was held out for testing and the remaining subjects formed the training set. All preprocessing and any resampling were fit on the training data only, and the learned parameters were applied to the held-out subject to prevent information leakage. Unless stated otherwise, RF hyperparameters were n_estimators = 500, max_features = “sqrt”, max_depth = None, min_samples_leaf = 1, and random_state = 42.

To address class imbalance, we compared three strategies within each training fold only: (i) no correction; (ii) class weighting (class_weight = “balanced” in scikit-learn), where the weight for class c was set to $w_{c}=\frac{N}{K*n_{c}}$ with N the number of training samples in the fold, K the number of classes (=3), and $n_{c}$ the count of class c in that fold; and (iii) uniform random oversampling with replacement by duplicating minority-class samples until an exactly balanced 1:1:1 ratio (Low/Medium/High) was reached. In each fold, the target per-class size equaled the majority-class count in that training fold. All resampling used random_state = 42 for reproducibility and was applied only to the training folds; the held-out subject’s data were never reweighted or resampled. No synthetic sample generation (e.g., SMOTE) was used.

For interpretability, we summarized impurity-based RF feature importances and examined their distribution after aggregating features by anatomical groups (e.g., trunk; hip–knee–ankle). Unless stated otherwise, importances were computed from the RF trained under the oversampling setting. Group contributions were computed by summing the importances of features associated with each segment within each fold and normalizing to 100%. We report accuracy and macro ROC AUC (one-vs-rest) as LOSO point estimates; confidence intervals are not provided because per-subject prediction logs were not preserved. Summary point estimates are provided in Table 3. Analyses were performed in Python (version 3.10) using standard, widely available machine-learning libraries (e.g., scikit-learn 1.3, NumPy 1.25, pandas 2.0). Exact package versions and hardware details were not recorded in this feasibility study. Scripts are available from the corresponding author upon reasonable request.

## 4. Results

### 4.1. Classification Performance

Under LOSO, Table 3 summarizes point estimates (accuracy and macro ROC AUC; one-vs-rest) across the three class imbalance settings. Without correction, overall accuracy was 0.61 with weak discrimination for the Medium class. Class weighting improved balance but remained below oversampling, which achieved the best performance (accuracy 0.86; macro ROC AUC 0.98). As per Methods (Section 3.5), we report LOSO point estimates only; confidence intervals are not provided. These results should be interpreted in light of the small, sex-imbalanced cohort, a single simple task, and CR10-only labeling (no talk test), as detailed in Section 5.

These results should be interpreted with several caveats. Performance depended on class imbalance handling—oversampling outperformed no correction or class weighting—and misclassifications occurred primarily between adjacent classes (Low vs. Medium and Medium vs. High), as seen in the confusion matrix. The values are LOSO point estimates without confidence intervals in a small (*n* = 10) cohort performing a single simple task from a single front-view camera; generalization to other tasks or viewpoints remains uncertain. In addition, depth-based skeleton tracking can produce short dropouts; although we smoothed/interpolated these (Section 3.4), such preprocessing may attenuate rapid, high-frequency changes.

### 4.2. ROC Analysis

The ROC curves for the three conditions are shown in Figure 4. The curve corresponding to oversampling clearly outperformed the other two, confirming the effectiveness of imbalance correction in improving classification reliability across Low, Medium, and High fatigue levels.

### 4.3. Confusion Matrices

Confusion matrices are presented in Figure 5. Without correction, frequent misclassification occurred between Low and Medium fatigue levels. Class weighting reduced this tendency, while oversampling yielded a more balanced and accurate classification across all three classes.

### 4.4. Feature Importance

The relative importance of kinematic features is shown in Figure 6 (see Table 2 for definitions). Importances were computed from the random forest trained with oversampling, were normalized within each fold to sum to 1, and were then averaged across folds. No single feature dominated; joint angles, angular velocities, and cycle timing each contributed modestly. This pattern indicates that fatigue is better captured by distributed changes across feature types and segments than by any isolated marker. For interpretability, we also summarize importances at the body-segment level, which corroborates this distributed pattern.

## 5. Discussion

This study demonstrated the feasibility of estimating subjective fatigue in healthy young adults using a non-contact approach with Azure Kinect and machine learning. A random forest classifier, trained on 24 kinematic features extracted from a simple whole-body task, achieved robust classification performance after imbalance correction (accuracy 0.86; macro ROC AUC 0.98, LOSO point estimate). These findings support the potential of Kinect-based systems as unobtrusive alternatives to wearable or physiological sensors for fatigue monitoring.

As described in Section 3.5, we computed impurity-based random forest feature importances and qualitatively inspected their distribution. To relate model behavior to anatomy, we grouped features into coarse anatomical regions (e.g., trunk; hip–knee–ankle; upper limb). This inspection suggested that importances were distributed across regions rather than concentrated in a single segment. In future work, we will apply permutation importance and SHAP to windowed features to quantify segment-wise, time-resolved contributions while keeping the non-contact pipeline unchanged.

Previous research on fatigue monitoring has often relied on wearable devices, such as heart rate sensors, EMG, or inertial measurement units [2,3,4,5,22,23,24]. While effective, these methods require direct attachment to the body, which may interfere with natural movement. Recent studies have explored vision-based sensing for motion and fatigue analysis, reporting encouraging results in controlled settings [8,9,10,11,12,13]. The present study extends this literature by demonstrating that even a simple, reproducible exercise performed by healthy individuals can yield reliable fatigue estimates from kinematic data alone.

Compared with prior work that often targets athlete or patient cohorts, specialized tasks, and binary fatigue labels using wearable signals (e.g., HR/HRV, EMG, IMUs) [2,3,4,5,22,23,24], camera-only approaches have shown feasibility in controlled settings [8,9,10,11,12,13] but typically adopt simpler label schemes or more complex pipelines. Table 4 provides a compact, cross-modal summary (camera-based and physiological) including modality, participants, task, labels, and the top metric (as reported), positioning our method as a camera-only reference baseline. While camera-only pipelines reduce burden, wearable or multimodal systems can capture cardio-respiratory or neuromuscular signals (e.g., HR/HRV, EMG) that may complement kinematic cues; benchmarking and fusion will be pursued in follow-up studies. Our study differs by (i) using a single commodity depth camera with off-the-shelf skeleton tracking, (ii) deliberately selecting a simple, reproducible whole-body movement that induces fatigue quickly and is reliably tracked, (iii) labeling three levels of subjective fatigue (CR10) without additional instrumentation, and (iv) explicitly evaluating class-imbalance handling under LOSO. These choices position our results as a camera-only reference baseline for non-contact deployments, clarifying when a camera-only setup can provide actionable fatigue cues relative to wearable or multimodal systems. For example, Aguirre et al. (sit-to-stand with Kinect + HR) [25] and Zhang et al. (IMU-based walking) [26] reported strong performance under different settings, complementing our camera-only setup.

Several limitations warrant mention. First, the cohort was small (*n* = 10; 8 male/2 female), limiting statistical power and generalizability; we will expand to a larger, sex-balanced cohort and, in that validation, explicitly examine whether anthropometrics (e.g., BMI and body mass) modulate kinematic patterns by quantifying their associations with kinematic feature distributions and classification performance. Second, we evaluated a single simple exercise with a single front-view depth camera; performance may vary with other tasks, speeds, viewpoints, or occlusion. Future work will extend beyond a single movement by including functional tasks (e.g., sit-to-stand, squats, walking/running) and testing whether the present feature set improves generalization across tasks or whether higher-capacity temporal/graph models do so. Third, labels relied solely on CR10; a talk test or physiological criteria were not administered. In follow-up validation, we will integrate a brief, standardized talk test (e.g., a 20–30 s passage) aligned with the labeling windows and report concordance with CR10 (e.g., Cohen’s κ and ROC analysis against a CR10 ≥ 7 threshold), together with HR/HRV and surface EMG as objective checks. Fourth, we report LOSO point estimates only because per-subject prediction logs were not preserved; to quantify uncertainty, future work will retain foldwise predictions and compute bootstrap confidence intervals for LOSO. Fifth, although oversampling improved class balance, it can bias performance estimates; we will validate on larger, sex-balanced datasets and explore cost-sensitive learning. Finally, the model scope was limited to a random forest; benchmarking higher-capacity sequence models (e.g., LSTM/GRU/temporal CNN/Transformers) and skeleton-graph networks (e.g., ST-GCN/GCN) is deferred to larger datasets, and we will extend interpretability with TreeSHAP, permutation importance, and PDP/ALE, including segment-level and cycle-phase-specific attributions.

From a deployment standpoint, microwearables and our camera-based, non-contact pipeline serve complementary needs and can be used alone or in combination. Our earlier note about motion interference pertains to conventional wearables; smart microwearables are skin-conformal and minimize such interference while providing access to physiological signals. Conversely, image-based sensing offers low setup burden and preserves natural movement; however, many microwearable platforms remain at the prototype stage with outstanding issues of cost and availability [6,7]. Depending on context, the two modalities may be deployed separately or jointly; benchmarking and fusion will be pursued in follow-up studies.

Future work will expand to larger and more heterogeneous populations, including older adults and clinical groups, and will explore integration with multimodal signals (e.g., HR/HRV, surface EMG, and microwearables) to improve accuracy and robustness [23]. Such developments could support personalized training, workplace ergonomics, and rehabilitation monitoring, where unobtrusive fatigue assessment is highly valuable.

Beyond research settings, the same non-contact pipeline could support (i) rehabilitation/physical therapy by aligning tasks with therapist-defined movements and flagging fatigue-related coordination changes during simple home or clinic exercises, with clinical oversight and objective checks (e.g., HR/HRV, surface EMG), and (ii) pets, where wearable attachment is challenging, by substituting an animal pose estimator (species-specific skeletal keypoints) and using veterinary-grounded labels. Both scenarios will require viewpoint-appropriate camera placement, occlusion handling, and broader, domain-specific datasets; model retraining and validation will be needed to address domain shift (assistive devices, altered ranges of motion, species differences).

## 6. Conclusions

This study demonstrated that subjective fatigue can be estimated in healthy young adults using a non-contact approach that combines Azure Kinect with machine learning. A random forest classifier trained on 24 kinematic features extracted from a simple, reproducible whole-body task achieved high performance after imbalance correction (macro ROC AUC = 0.98, LOSO point estimate). Feature importance analysis revealed that contributions from multiple joint motions were more informative than any single marker, underscoring the value of distributed kinematic changes in fatigue estimation.

By focusing on a small, homogeneous sample and a single movement task, this study clearly established the feasibility of the proposed approach under controlled conditions. These deliberate choices highlight the method’s strength in yielding reliable results from minimal experimental requirements. Building on this foundation, future work will expand to larger and more diverse populations, incorporate multiple exercise tasks, and integrate multimodal signals to further enhance accuracy and robustness. Such developments could pave the way for widespread applications in personalized training, workplace ergonomics, and rehabilitation monitoring, where unobtrusive fatigue assessment is particularly valuable. Taken together, the proposed non-contact approach demonstrates feasibility and points toward clinical rehabilitation (with objective checks and clinical oversight) and pet monitoring (via animal pose estimators and veterinary-grounded labels), alongside multimodal fusion with microwearables and richer interpretability (e.g., class-wise attributions and cycle-phase-specific analyses) as key next steps.

## Figures and Tables

**Figure 1 sensors-25-06633-f001:**
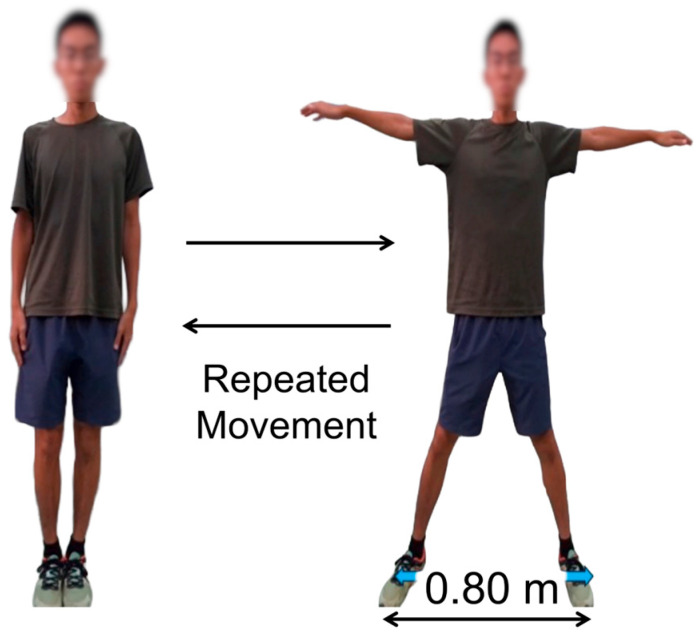
Repeated whole-body movement used for data collection: participants alternated between a neutral stance and an arms-abducted posture at a self-selected pace; stance width was approximately 0.80 m.

**Figure 2 sensors-25-06633-f002:**
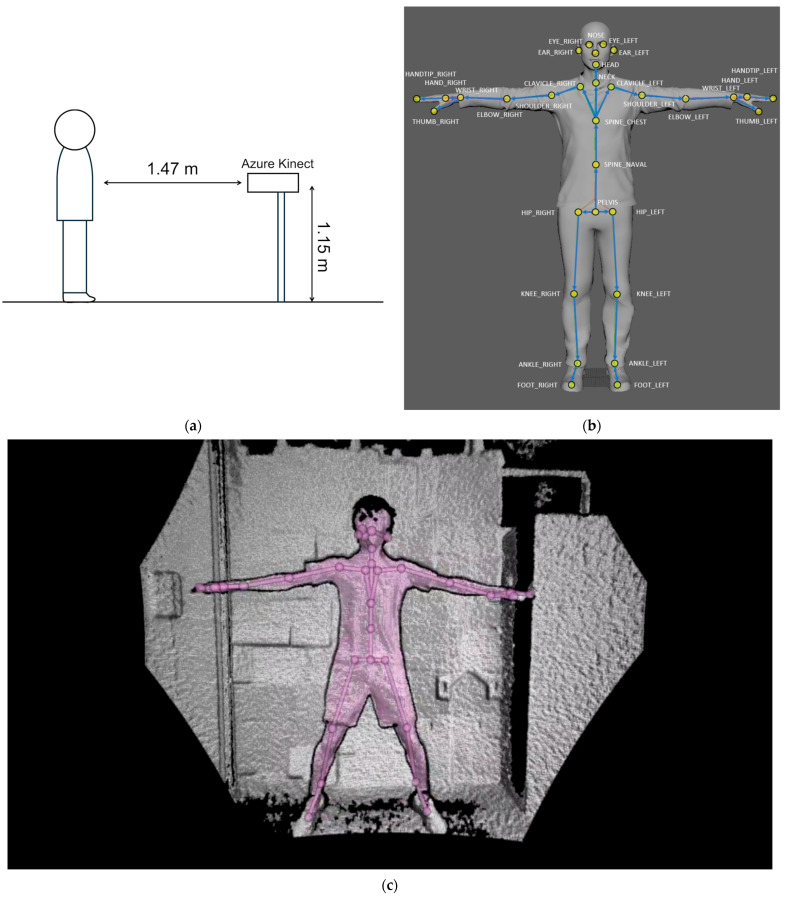
RGB-D capture setup and body-tracking reference: (**a**) measurement setup—Azure Kinect DK positioned 1.47 m in front of the participant at a height of 1.15 m (frontal view); (**b**) joint hierarchy defined by the Azure Kinect Body Tracking SDK (32 joints); (**c**) example frame from the study task with the 32-joint skeleton overlaid on the depth image (circles = joints; lines = bones). Panel (**b**) reproduced from Microsoft Learn, “Body joints—Joint hierarchy,” © Microsoft, licensed under the Creative Commons Attribution 4.0 International (CC BY 4.0) license [21]. See Appendix A for a real-time demonstration presented sequentially—RGB, then depth, then depth with the 32-joint skeleton overlay.

**Figure 3 sensors-25-06633-f003:**
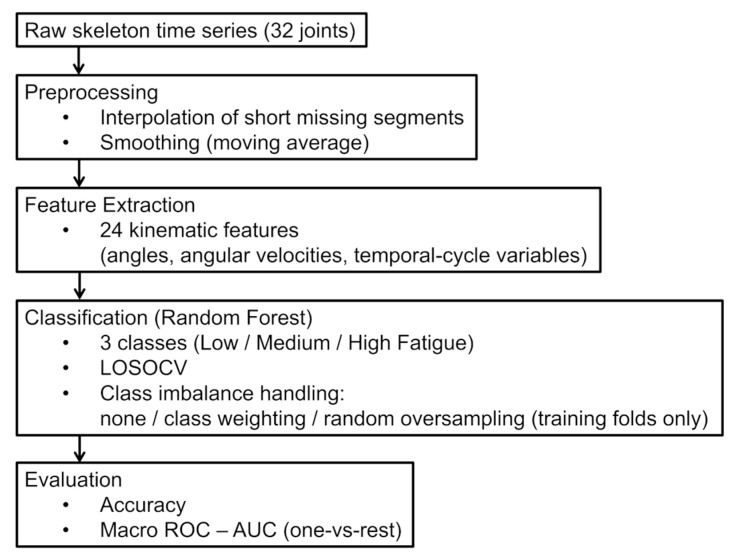
Processing pipeline: RGB-D capture and body tracking → preprocessing (interpolation, smoothing) → 24 kinematic features (angles, angular velocities, cycle timing) → random forest classification under LOSO → evaluation (accuracy, macro ROC AUC; one-vs-rest).

**Figure 4 sensors-25-06633-f004:**
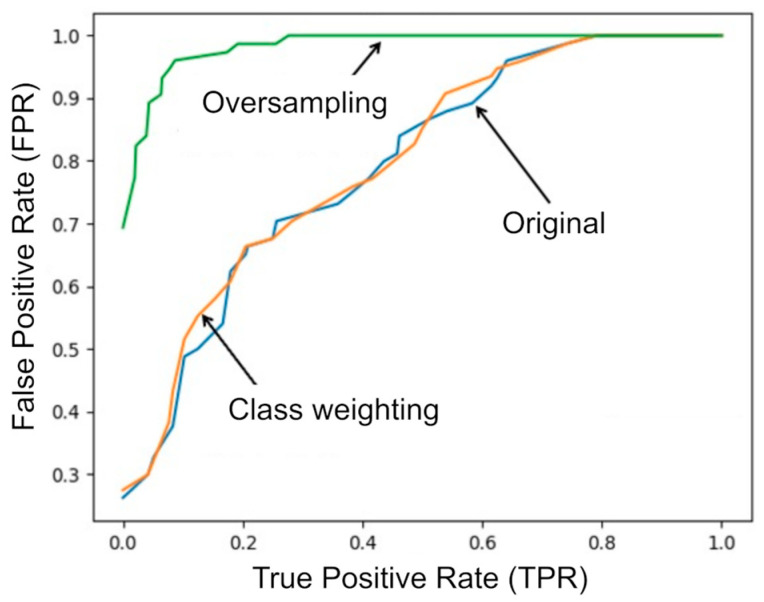
ROC curves of the random forest classifier under three imbalance correction strategies: original (no correction), class weighting, and oversampling.

**Figure 5 sensors-25-06633-f005:**
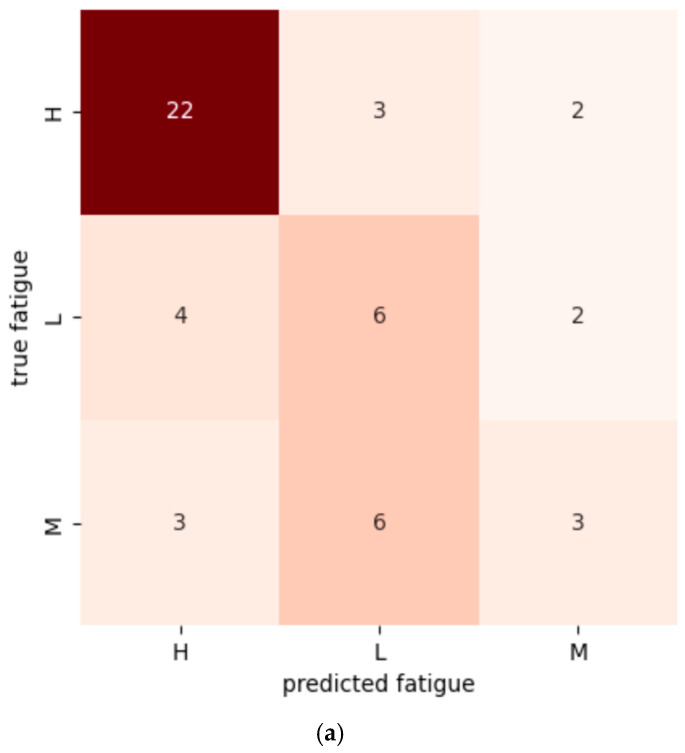
Confusion matrices of the random forest classifier: (**a**) no correction, (**b**) class weighting, (**c**) oversampling. Cell values are counts; color intensity corresponds to the count (darker = larger).

**Figure 6 sensors-25-06633-f006:**
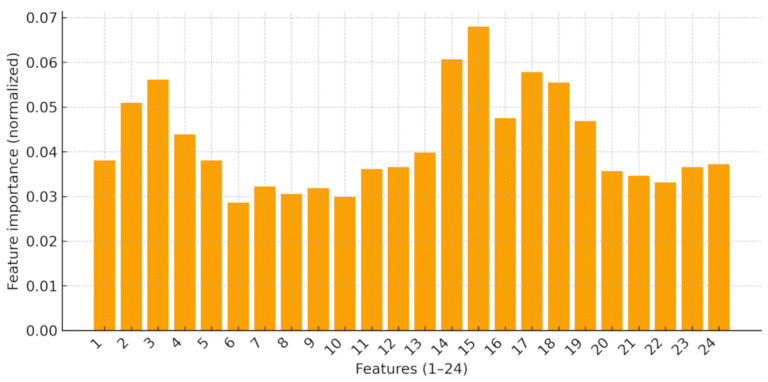
Feature importance of the 24 kinematic features (see Table 2 for feature definitions) after oversampling, calculated using the random forest classifier.

**Table 1 sensors-25-06633-t001:** Participant characteristics. Values are mean ± SD; *n* indicates the number of available records (ranges in parentheses).

Category	Value
Participants (*n*)	10
Sex	8 male, 2 female
Age (y), mean ± SD	18.5 ± 0.8 (17–19)
Height ^1^ (cm), mean ± SD	166.2 ± 9.5 (153.0–178.0)
Mass ^1^ (kg), mean ± SD	54.8 ± 7.0 (46.7–64.0)
BMI ^2^ (kg/m^2^), mean ± SD	19.8 ± 1.3 (17.4–22.4)

Note: ^1^ Height and mass were self-reported during revision; only anonymized aggregate statistics are reported. Note: ^2^ BMI = body mass index (kg/m^2^), computed as mass (kg)/[height (m)]^2^.

**Table 2 sensors-25-06633-t002:** Kinematic features * used for classification (Features 1–24).

Feature ID	Description
1	Mean hip–knee opening angle (left leg, close → open)
2	Mean hip–knee opening angle (right leg, close → open)
3	Mean shoulder abduction angle (left arm, close → open)
4	Mean shoulder abduction angle (right arm, close → open)
5	Peak angular velocity (left leg, close → open)
6	Peak angular velocity (right leg, close → open)
7	Peak angular velocity (left arm, close → open)
8	Peak angular velocity (right arm, close → open)
9	Cycle time (left leg, close → open)
10	Cycle time (right leg, close → open)
11	Cycle time (left arm, close → open)
12	Cycle time (right arm, close → open)
13	Minimum angular velocity (left leg, open → close)
14	Minimum angular velocity (right leg, open → close)
15	Minimum angular velocity (left arm, open → close)
16	Minimum angular velocity (right arm, open → close)
17	Cycle time (left leg, open → close)
18	Cycle time (right leg, open → close)
19	Cycle time (left arm, open → close)
20	Cycle time (right arm, open → close)
21	Inter-cycle interval (left leg, open → open)
22	Inter-cycle interval (right leg, open → open)
23	Inter-cycle interval (left arm, open → open)
24	Inter-cycle interval (right arm, open → open)

* Kinematic features are grouped into three categories—angles, angular velocities, and temporal-cycle variables. Angles and velocities were computed from joint coordinates converted from millimeters to meters; time series were smoothed before differentiation.

**Table 3 sensors-25-06633-t003:** Classification performance of the random forest under three imbalance-handling strategies. Metrics: accuracy and macro ROC AUC (one-vs-rest). Oversampling was applied within training folds only; the held-out subject’s data were never reweighted or resampled.

Condition	Accuracy	Macro ROC AUC
No correction	0.61	0.71
Class weighting	0.73	0.85
Oversampling	0.86	0.98

**Table 4 sensors-25-06633-t004:** Compact comparison of representative fatigue-estimation studies across modalities. Rows list studies; columns summarize modality, participants, task, labels, and the top metric (as reported) ^1^.

Study (Year)	Modality	Participants	Task/Setting	Labels (Classes)	Top Metric (As Reported)
This work	Non-contact depth camera (Azure Kinect; skeleton tracking)	*n* = 10, healthy young adults	Simple reproducible whole-body movement	Borg CR10 (3: Low/Med/High)	Accuracy 0.86; macro ROC AUC 0.98
Aguirre et al., Sensors 2021 [25]	Kinect V2 + wearable HR (Zephyr)	*n* = 60 healthy volunteers	Sit-to-stand, 120 s; stop if HR > 90% MHR or Borg = 10	Three fatigue conditions (low/moderate/high)	Accuracy 82.5%
Zhang, Lockhart & Soangra, Ann. Biomed. Eng. 2014 [26]	IMU (lower-limb) + SVM	*n* = 17 (29 ± 11 y)	Walking; fatigue induced by squats to 60% baseline MVE ^2^	Binary: fatigue vs. no-fatigue	Accuracy 96%

^1^ Metrics are reported as in the original sources; cohorts, labeling criteria, window definitions, and validation protocols (e.g., leave-one-subject-out [LOSO] vs. random split) differ across studies; therefore, values are not directly comparable. ^2^ Fatigue was induced to 60% of baseline MVE (maximum voluntary exertion) before the walking trials.

## Data Availability

The data presented in this study are available on request from the corresponding author. The data are not publicly available due to privacy restrictions.

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
