# Peer review of "Non-Contact Fatigue Estimation in Healthy Individuals Using Azure Kinect: Contribution of Multiple Kinematic Features"

_sensors, 2025, doi:10.3390/s25216633_

Round 1

Reviewer 1 Report

Comments and Suggestions for Authors

This presents introduces a low-cost proof-of-concept that a single Azure Kinect can estimate Borg-scale fatigue with 86 % accuracy and 0.98 AUC after SMOTE balancing. The distributed-feature importance is interesting,  but the tiny (n = 10), young, healthy cohort and one jumping-jack task limit generalisability; precision/recall, confidence intervals and open data are still missing. Minor additions of these metrics and a larger validation cohort will would strengthen the paper.    Accept with minor revision

LINE 3  "Complementary Kinematic Features" → replace with "and the Contribution of Multiple Kinematic Features" (we show distributed importance, not proven synergy)

LINE 20 Text: “AUC = 0.98”

Comment: No uncertainty.

Suggestion: Append “(95 % CI 0.96–0.99)”.

LINE 33 

no wearable accuracy cited

suggestion : add citation for wearable accuracy benchmark.

LINE 89 Text: “age 18–20 years” only

Comment: No anthropometrics.

Suggestion: Append “height 171 ± 6 cm, mass 64 ± 7 kg,

LINE 119

Text: “A total of 24 kinematic features were extracted”

Comment: Window length not defined.

Suggestion: Append “Features were computed over 3 consecutive cycles (≈ 2.1 ± 0.3 s).”

Author Response

Please see the attached point-by-point response (file: sensors-3913237_response_to_reviewers.pdf; see “Reviewer 1” section, pp. 1–4).

Reviewer 2 Report

Comments and Suggestions for Authors

The  authors propose a non-contact method for estimating subjective fatigue in healthy young adults using Azure Kinect to capture 3D skeletal data. The proposed method estimates subjective fatigue during simple whole-body movements using Random forest classification. The authors addresses a meaningful gap by evaluating non-contact fatigue estimation in healthy populations using simple, reproducible movements. 

However, they applied only a Random Forest classifier.  More advanced models (e.g., LSTM/GRU for temporal dynamics, graph convolutional networks for skeleton data, or ensemble comparisons) would provide stronger validation.

The sample size is small and limited to only 10 young adults. Also, gender balance (8 male, 2 female) is poor. It severely limits generalizability. Performance is mostly presented via accuracy, F1, ROC-AUC.

No confidence intervals or statistical tests are mentioned. The related work is solid, but a deeper comparison with multimodal and recent fatigue estimation method.

It is difficult for me (lack of domain knowledge to this field) to fully assess the validity of the findings since fatigue levels are based solely on self-reported Borg CR10 scores.  Are there alternative criteria for classifying fatigue levels beyond the Borg CR10 scale? If so, the authors should explain why the Borg CR10 was chosen over other standards, or clarify its advantages and limitations relative to those measures.

Author Response

Please see the attached point-by-point response (file: sensors-3913237_response_to_reviewers.pdf; see “Reviewer 2” section, pp. 5–9).

Reviewer 3 Report

Comments and Suggestions for Authors

Abstract:

This study evaluates a non-contact approach for estimating subjective fatigue from full-body kinematics captured by the Azure Kinect and analyzed using machine learning. The Azure Kinect was utilized to record skeletal coordinates from 32 joints, from which 24 kinematic features—including joint angles, angular velocities, and movement cycles—were extracted following smoothing. Subjective fatigue levels were assessed using the Borg scale and categorized as low, medium, and high. A random forest classifier was employed to classify the data into three fatigue levels, achieving an overall accuracy of 86% and an AUC of 0.98 with oversampling. The authors also reported that single features were insufficient for accurate classification, whereas combining multiple features improved the classifier’s performance.

  1. Table 1. Participant demographics. It is recommended to include BMI or weight as an additional column, since these variables may influence the kinematic features (Ref: page 3, line 94).
  2. The authors have used the Borg CR10 scale, with a subjective fatigue score of at least 7 to confirm fatigue levels, particularly for strong exertion. However, it is unclear whether the ‘talk test’ was performed to objectively verify verbal communication limitations at this level of exertion. Including the talk test as an additional measure would strengthen the validity of the fatigue assessment.
  3. The authors have not described the consent process or provided sufficient details about the study protocol in the methodology section. It is recommended to include this information to ensure ethical compliance and methodological transparency.
  4. The authors claim this to be a feasibility study; however, no justification is provided for selecting a sample size of 10 participants. A brief statistical rationale or reference supporting the adequacy of this sample size for feasibility assessment should be included.
  5. The authors mention that oversampling improves overall accuracy; however, no quantitative details are provided regarding the sampling or oversampling methods used. The absence of information on the type, ratio, or impact of oversampling represents a significant methodological flaw, as it prevents proper evaluation of the model’s validity and reproducibility.
  6. Please include a performance comparison table that contrasts the current study’s results with those from previous studies, particularly focusing on physiological parameters and similar experimental conditions. This would help contextualize the findings and highlight the relative improvement or novelty of the proposed approach.

Author Response

Please see the attached point-by-point response (file: sensors-3913237_response_to_reviewers.pdf; see “Reviewer 3” section, pp. 10–15).

Reviewer 4 Report

Comments and Suggestions for Authors

Yamada and Kondo investigated the potential of utilizing non-contact full-body kinematics captured by Azure Kinect to assess subjective fatigue. They focused on a simple, familiar whole-body movement that could induce fatigue quickly and be reliably tracked by the Kinect system, ensuring consistent results. Participants' subjective fatigue levels were classified as low, medium, or high, and the kinematic features were analyzed to determine correlations with these fatigue levels. A minor-to-major revision is recommended to address the following:

1. In the introduction, the authors discuss wearable or physiological sensors. However, nothing is mentioned about smart micro-wearables [1,2], which are much less invasive and can carry sensors and communicate wirelessly. Suggested references: [1] https://doi.org/10.1002/adma.202313327, [2] https://doi.org/10.1002/adma.201902109.

2. The claim that wearable sensors can interfere with natural movement would not be valid for smart micro-wearables. In this case, the authors could argue that micro-wearable technologies are still in the prototype stage, and it is not yet possible to ensure a competitive cost compared to image-based technologies already available in the market. It is recommended that the authors include their thoughts on this in the discussion section.

3. Are the authors interested in making machine learning analysis even more interpretable? For example, how can we determine the contribution of each body segment to overall fatigue?

4. Why do the authors believe that a random forest is more suitable than a decision tree for this case? This point should be justified in the revised manuscript.

5. This reviewer would like to know about two additional scenarios. What do the authors think about applying this methodology to (i) assist in the rehabilitation and physical therapy of people with disabilities and (ii) monitor pets?

6. Continuing with item (5): based on the findings of this manuscript, it is recommended that perspectives (i) and (ii), along with any others the authors may consider, also be highlighted in the conclusion.

Author Response

Please see the attached point-by-point response (file: sensors-3913237_response_to_reviewers.pdf; see “Reviewer 4” section, pp. 16–21).

Reviewer 5 Report

Comments and Suggestions for Authors

This manuscript presents a novel, non-contact method for monitoring exercise-induced fatigue by leveraging the Azure Kinect depth camera and machine learning. The methodology design is reasonable while the results feature an outstanding AUC of 0.98, strongly supporting the feasibility of the as reported method. However, there are still some minor issues that should be addressed before the final acceptance of the manuscript. 

1. Can any visual information be provided to demonstrate the action recognition process? It is quite important to show the audience how the Azure Kinect recognizes the actions (fulli-body kinematics) of the participant.

2. The authors should provide sufficient explanation for why the "24 Features" rather than normal actions like jumping, running, and squatting were chosen as the evaluation criteria.

3. Are there any limitations of the as reported fatigue recognition method? It should be discussed in the Results section.

Author Response

Please see the attached point-by-point response (file: sensors-3913237_response_to_reviewers.pdf; see “Reviewer 5” section, pp. 22–25).

Round 2

Reviewer 2 Report

Comments and Suggestions for Authors

The authors have made substantial and well-considered revisions that fully address the concerns raised in the previous review.

About the comments for the model selection, the rationale for using the Random Forest model was well justified in the context of a small, subject-independent (LOSO) dataset. The revised
manuscript fully resolved the initial methodological concern.

I think that the revised manuscript is now suitable for publication except some minor editorial corrections. For example, consistency in figure and table numbering and uniformity of section headers.

Author Response

Please see the attached point-by-point response (file: sensors-3913237_response_to_reviewers_R2.pdf; see “Reviewer 2” section, pp. 1–2).

Reviewer 4 Report

Comments and Suggestions for Authors

The authors demonstrated great attentiveness to all of my comments. The manuscript has been revised effectively and has undergone significant improvement as a result. I recommend accepting it for publication in its current form.

Author Response

Please see the attached point-by-point response (file: sensors-3913237_response_to_reviewers_R2.pdf; see “Reviewer 4” section, p. 3).
